# Wound Healing, Anti-Inflammatory and Anti-Oxidant Activities, and Chemical Composition of Korean Propolis from Different Sources

**DOI:** 10.3390/ijms252111352

**Published:** 2024-10-22

**Authors:** Aman Dekebo, Chalshisa Geba, Daniel Bisrat, Jin Boo Jeong, Chuleui Jung

**Affiliations:** 1Department of Applied Chemistry, Adama Science and Technology University, Adama 1888, Ethiopia; amandekeb@gmail.com; 2Agricultural Science and Technology Research Institute, Andong National University, Andong 36729, Republic of Korea; danielbisrat@gmail.com; 3Department of Pharmacognosy and Pharmaceutical Chemistry, Addis Ababa University, Addis Ababa 1176, Ethiopia; chaligbegna4@gmail.com; 4Department of Forest Science, Andong National University, Andong 36729, Republic of Korea; jjb0403@anu.ac.kr; 5Department of Plant Medicals, Andong National University, Andong 36729, Republic of Korea

**Keywords:** flavonoids, pinocembrin, chrysin, antioxidant, anti-inflammatory, Korean propolis

## Abstract

Propolis, such as is used as bio-cosmetics and in functional materials, is increasing because of its antioxidant medicinal benefits. However, its pharmacological and chemical composition is highly variable, relative to its geography and botanical origins. Comparative studies on three propolis samples collected from different regions in Korea have been essential for linking its provenance, chemical composition, and biological activity, thereby ensuring the efficient utilization of its beneficial properties. Here, we report the chemical composition and biological activities such as the antioxidant, wound healing, and anti-inflammatory effects of the ethanolic extract of Korean propolis collected from two regions. We compared the chemical constituents of three 70% ethanol-extracted (EE) samples, including the Andong, Gongju field (GF), and Gongju mountain (GM)-sourced propolis using Gas chromatography–mass spectrometry (GC–MS). The major and common components of these EE Korean propolis were flavonoids such as pinocembrin (12.0–17.7%), chrysin (5.2–6.8%), and apigenin (5.30–5.84%). The antioxidant property using a 2,2-Diphenyl-1-picrylhydrazyl (DPPH) radical scavenging activity assay of EEP showed substantial differences among samples with the highest from Andong. The sample 10% GM levigated in simple ointment was found to be the most active in wound healing activity based on the excision, incision, and dead space wound models. The potential of propolis for wound healing is mainly due to its evidenced properties, such as its antimicrobial, anti-inflammatory, analgesic, and angiogenesis promoter effects, which need further study. The anti-inflammatory activity and NO production inhibitory effect were highest in GM samples. However, GM and GF samples demonstrated similar inhibitory effects on the expression of inflammatory mediators such as iNOS, IL-1β, and IL-6. The presence of a higher concentration of flavonoids in Korean EE propolis might be responsible for their promising wound healing, anti-inflammatory, and antioxidant properties.

## 1. Introduction

Wound healing is a complex tissue repair process that restores damaged and lost cellular structures and tissue layers. It takes place in four well-coordinated phases: blood clotting (hemostasis), inflammation, tissue formation (proliferation) involving both parenchymal and connective tissue cells and the synthesis of extracellular matrix proteins, and tissue remodeling (maturation). This leads to the formation of scar tissue, accompanied by wound contraction and epithelialization [1,2]. Various factors, such as age, sex, nutrition, stress, infections, and medications, can disrupt one or more of these phases, potentially resulting in delayed or impaired healing [3,4]. Wound defects can be managed through various approaches. For small, uncomplicated wounds, wound coverage is the optimal treatment. However, managing more complex wounds such as diabetic, chronic, infected, burned, or large wounds leads to technical challenges [5,6]. Healing in these cases is more complicated, and beyond preventing and eliminating infection, the goal is to promote wound contraction and epithelialization while controlling scar tissue formation to prevent excessive scarring and contracture [6].

Natural products such as propolis have long been popular in alternative healing therapies due to their therapeutic properties, availability, and affordability. Since ancient times, propolis has been regarded by many as a remedy for skin diseases. By the late 19th century, propolis was recognized as a wound healing agent, and it was used in the Second World War in the Soviet Union [7].

Propolis is a material in the hive collected by honeybees from different plant parts, such as buds and exudates, and mixed with bee saliva [8]. Honey bees such as *Apis mellifera* and *A. cerana* use propolis to seal hive walls and its entrances to strengthen the combs and embalm dead invaders [9]. Propolis has been utilized as a traditional medicinal substance valued for internal and external uses since ancient times in many countries around the world [10]. Egyptians, Greeks, and Romans reported using propolis for its general healing qualities, with ancient Egyptians specifically utilizing it for embalming their dead [3]. Nowadays, it has gained a lot of acceptance as medicinal treatments in various regions is claimed to promote tissue repair, accelerate wound contraction, shorten healing time, and reduce scar formation after wound healing and diseases including diabetes and cancer [11,12,13]. Currently, propolis is widely used as liquid extracts, vaporizers, syrup capsules, tablets, candies, creams, etc., in different countries [14,15].

Depending on the source plants, the color of propolis varies from yellow to dark brown [8,9]. Previously, several biological activities including the acceleration of regenerating processes in the damaged cartilages and bones [16,17] and immunomodulatory [18], antifungal [10,19,20], antiviral [21], analgesic [22], anti-inflammatory [23], antitumoral [24], antioxidant [25], antibacterial [19,20,26] activities, etc., were reported for different extracts and compounds of propolis, which has led to increased interest in these materials in food and cosmetic industries.

Various chemical investigations on propolis solvent extracts and its essential oils showed that its composition is complex and depends on the local plant flora, regions, and climatic conditions where the sample was collected [27]. Many important chemical classes of compounds such as flavonoid, phenolic, and aromatic compounds [21] have been identified in propolis samples [27]. Flavonoids are considered the principal compounds responsible for the beneficial effects of propolis. These plant-derived phenolic compounds play a role in various physiological processes, including vitamin absorption, wound healing as antioxidants, and exerting antimicrobial effects, while also modulating the immune system [28].

The biological activities of propolis vary due to differences in its chemical composition, which in turn depend on the local plant flora, regional variations, and climatic conditions, even within the same country. There are some studies about the anti-inflammatory and antioxidant effects of Korean propolis [29,30,31,32]. However, to the best of our understanding, there were no further studies such as the comparison of chemical composition, wound healing, and anti-inflammatory and antioxidant effects on propolis collected in different regions in South Korea. Therefore, this study aimed to analyze and compare the chemical composition of Korean propolis, as well as evaluate the wound healing, anti-inflammatory, and antioxidant properties of ethanolic extracts of Korean propolis, obtained from three different localities.

## 2. Results and Discussion

### 2.1. Extract Yield

The ground propolis samples (50 g) were extracted in 95% *v*/*v* ethyl alcohol, yielding ethanolic crude extracts of propolis from Gongju mountain, Gongju field, and Andong hives as follows: 23.5 g (47.0%), 23.8 g (47.6%), and 23 g (46.0%), respectively. The percentage yield of the extracts from the three sites is comparable. The ethanolic extract of the propolis of Korean origin (EEP-K) percentage yields obtained in this study was comparable to those reported by Uzel et al. [33] (20.5–44.8%) and higher than those reported by Markiewicz-Żukowska et al. [34], which was 16.3%. The difference in the percentage yield of the extract depends on the geographical region, vegetation types of its sites, and extraction methods.

### 2.2. Thin-Layer Chromatography (TLC) Analysis

The phytochemical composition comparison of the ethanolic extract of the propolis (EEP) of the three samples was carried out using TLC as described in the Materials and Methods Section. All three EEP samples exhibit similar TLC profiles (Figure 1), consistent with those reported to originate from *Populus* spp. (poplar) [35]. Dark TLC spots at R_f_ = 0.43 and 0.66 under UV 254 nm and a white colored spot at R_f_ = 0.03, light blue spot at R_f_ = 0.34, dark blue spot at R_f_ = 0.60, and light-yellow spot at R_f_ = 0.64 under UV 366 nm are commonly found in Korean propolis and might be used as makers. Thus, the analyzed Korean samples originated from poplar bud. Bankova et al. [36] reviewed propolis from Europe and Asia that originated from *Populus* spp. (poplar) and constituted pinocembrin, pinobanksin, and pinobanksin-3-O-acetate, as well as those from North America, constituting chrysin, galangin, and caffeates (benzyl, phenylethyl, and prenyl).

### 2.3. Chemical Composition

The ethanolic extract of Korean propolis chemical composition was shown in Table 1 and Figure 2. A total of 73 compounds were identified from Korean propolis samples. The chemical composition of propolis depends on the vegetation types near the apiaries [37]. It was reported that propolis samples from temperate zone regions such as Asia, Europe, and North America contain mainly phenolic compounds, such as flavonoids, aromatic acids, and their esters. The plant source is the poplar bud (*Populus* spp.) as it is dominant tree species available as a source of propolis [38]. In our study on three Korean propolis samples, the predominant and common constituents were flavonoids such as pinocembrin (12.00–17.71%), chrysin (5.20–6.75%), and apigenin (5.30–5.84%) aromatic acid esters such as methyl 3,4-dimethoxycinnamate (3.90–7.76%), benzyl alcohol (3.40–9.79%), pinostrobin chalcone (2.67–10.68%), and 5-hydroxy-7-methoxyflavone (3.30–5.44%) (Figure 3, Table 1). Our results are in good agreement with regard to the presence of flavonoid and aromatic acid ester compositions of propolis with those reported by other authors [39] in Bulgarian propolis samples. Four Anatolian propolis samples were also reported to have principal flavonoids [33].

Chrysin was reported as a propolis characteristic compound, and its highest percentage compositions were quantified in propolis from Turkey and Uruguay (>2000 μg/g) compared to those of Polish origin (>1000 μg/g) [40]. Pinocembrin identified in propolis in our study was also reported as a major compound in propolis samples from Turkey and Uruguay (1726.058–2816.289 μg/g) [40]. 

A previous study also indicated caffeic acid phenethyl ester and chrysin as active constituents of propolis. It also indicated that they inhibit the growth of the astroglia cell line (SVGp12) by activating a cytotoxic effect [34]. Pinostrobin chalcone identified in all propolis samples of Korean origin in this study was reported to exhibit cytotoxic activities against MDA-MB-231 and HT-29 colon cancer cell lines [41].

Granados-Pineda et al. [42] isolated pinocembrin, pinobanksin, chrysin isorhamnetin, pinobanksin-5-methylether alpinetin, alpinone, pinostrobin, galangin-5-methylether, and kaempferide from brown propolis samples collected from Chihuahua, Durango, and Zacatecas in Mexico. Pinocembrin was reported to be common in the three Mexico propolis samples and had the highest yield. Additionally, these workers reported that pinocembrin is used to treat diabetic nephropathy when there is no kidney damage.

### 2.4. Antioxidant Activity

The antioxidant activity of propolis extracts was examined by the DPPH radical scavenging method. As shown in Figure 4, DPPH radical scavenging activity was significantly highest (*p* < 0.05) for Andong propolis extract compared with Gongju mountain and Gongju field propolis extracts. The Gongju mountain extracts and Gongju field propolis extracts have comparable antioxidant activities, and statistically there is no significant difference among them.

Choi et al. [43] evaluated the antioxidant activity of different propolis extracts in Korea, compared them with Brazilian propolis samples, and reported that propolis from Korea had stronger DPPH free radical scavenging activity. Choi et al. [44] investigated the antioxidant effects and phenolic constituents of Korean propolis collected from different locations such as Uijeongbu, Ansan, Hongcheon, Youngwol, Chungju, Cheongju, Cheonan, Daejeon, Iksan, Jangseong, Kwangju, Sangju, Gumi, Changnyeong, Uiryeong, Aewol, and Pyoseon. They reported that all samples of propolis, except for those from Hongcheon, Iksan, Aewol, and Pyoseon, had relatively strong antioxidant properties and high total polyphenol contents. However, there was no previous report on the anti-oxidant activity of propolis samples collected from Andong and Gongju, South Korea. Socha et al. [45] also studied phenolic acid and flavonoid contents, as well as the antioxidant properties of propolis samples collected from different regions of Poland, and reported that the samples exhibited antioxidant activities correlated with their total flavonoid contents [45].

### 2.5. Acute Dermal Toxicity 

In the assessment of acute dermal toxicity, the maximum concentration of the ointment (10% *w*/*w*) applied at a limit dose of 2000 mg/kg was found to be safe. No signs of salivation, tremors, convulsions, diarrhea, lethargy, coma, or death were observed 24 h after application to the shaved dorsal area. Similarly, no such symptoms were noted during the 6 h and 48 h monitoring periods. Furthermore, there were no signs of toxicity or mortality observed throughout the 15-day observation period.

### 2.6. Wound Healing Activity

#### 2.6.1. Excision Model

##### Wound Contraction

The daily healing development of wound contraction produced by the topical application of 5% and 10% *w*/*w* ointment of ethanolic propolis extract is shown in Table 2 and Appendix A. The mean of the wound creation area in the mm^2^ (day 0 in the mm^2^ wound area) of each treated group are SO = 316.16, 5% AP = 312.44, 10% AP = 316.62, 5% GFP = 309.32, 10% GFP = 308.22, 5% GMP = 307.48, 10% GMP = 306.66, and 0.2% NF = 312.40 mm^2^. The percentage wound closure for each group was calculated from these data using the following formula.
$$\%Wound\;contraction=\frac{Wound\;area\;on\;day\;0\;–\;Wound\;area\;on\;day\;n}{Wound\;area\;on\;day\;0}\times 100$$
where n = days of wound contraction measurement consecutively on 2nd, 4th, 6th … 20th days.

The wound contraction activity of the ethanolic propolis extracts increases in a dose-dependent manner. The Gongju mountain ethanolic propolis extract and 0.2% nitrofurazone used as the positive control standard drug exhibited a similar onset of action in reducing the wound size. This was significant (*p* < 0.001) compared to the simple ointment treatment group in all days observed. This extract has a wound contraction effect at the lowest concentration tested (5%), which is significantly different from the negative control. Five percent Andong ethanolic propolis extract showed the lowest wound contraction effect. There was no wound area on day 18 for both 10% GMP and 0.2% standard drug. Similar wound contraction action was observed for 5%, 10%, and 5% GFP, GFP, and GMP, respectively, on day 20. The effect of the ethanolic extracts of propolis in a percentage of the wound closure excision model in mice was consistence with those of the wound contraction area. Ten percent GMP exhibited the highest percentage of wound closure, which was similar to that of the standard drug. A percentage of wound closures showed the order NF > 10% GMP > 5% GMP > 10% GFP > 5% GFP > 10% AP > 5% AP > SO (Table 3, Figure 5).

##### Epithelialization Period (EP)

The EP of all formulations tested in excision wounds is indicated in Table 4. The time required to completely heal the wound was significantly (*p* < 0.01) shortest in rats treated with extract ointments 10% GMP (15.11 ± 0.44 days), which is not significantly different from that of 0.2% nitrofurazone (14.92 ± 0.64 days) and different from those of simple ointment (SO). On average, the period of epithelialization was 14.92 ± 0.64, 15.11 ± 0.44, 16.43 ± 0.22, 16 ± 0.18, and 16.43 ± 0.12 for 0.2% NF, 10% GMP, 5% GMP, 10% GFP, and 5% GFP, respectively.

#### 2.6.2. Incision Model

In the wound healing process, the deposition of newly synthesized collagens at the wound site increases collagen concentration per unit area and, hence, the tissue tensile strength [46]. Table 5 compares the tensile strength of the healing skin treated with different propolis ethanolic extract formulations for 20 days. Minimum tensile strength was observed for the untreated control (178.18 ± 6.55) followed by those treated with simple ointment (206.56 ± 7.75). The tensile strength of the tissue treated with other propolis ethanolic extract formulations was significantly higher in the treated group than the negative control group. The tensile strength of the wound treated with 10% GMP propolis ethanolic extract formulation was the highest (338.13 ± 3.48), which was not statistically different from the 0.2% nitrofurazone ointment (352.76 ± 5.66) that was used as the positive control. The percentage of tensile strength calculated for 10% (*w*/*w*) GMP was 63.82%. Even 5% (*w*/*w*) AP, which resulted in the weakest tensile strength compared with other extracts, was significantly different from the negative control. In general, collagen synthesis and strength by the formation of inter- and intra-molecular cross links was increased with increased tensile strength, and this process facilitated wound healing [47].

#### 2.6.3. Estimation of Hydroxyproline Content 

Wound healing primarily depends on the tissue’s repair capacity, the type and extent of the injury, and the overall health of the tissue. The granulation tissue in a wound consists mainly of fibroblasts, collagen, and small, newly formed blood vessels. Collagen, which is rich in the amino acid hydroxyproline, is the key component of the extracellular matrix, providing structural strength and support. The breakdown of collagen releases free hydroxyproline and its peptides [48]. Therefore, measuring hydroxyproline in granulation tissue can provide insight into the maturation and healing process [49]. As shown in Table 6 and Appendix A, the hydroxyproline content in the granulation tissue of animals treated with 10% GMP and 5% GMP ethanolic propolis extracts were significantly higher compared to other groups.

Wound healing is a complex process that includes several mechanisms such as hemostasis, inflammation, matrix synthesis and deposition, angiogenesis, fibroplasia, epithelialization, contraction, and remodeling. Any disruption in these processes can extend tissue damage and delay recovery. In this context, various types of propolis have been found to promote faster wound healing by acting on different stages of the healing process [50,51,52,53]. Propolis has been shown to minimize healing duration, increase wound contraction, facilitate epithelialization, and speed up tissue repair processes [2,53,54]. It has also been reported that the topical application of propolis can reduce mast cell counts, thereby accelerating the healing process, making it highly promising for wound healing. Propolis significantly reduces mast cell counts in surgical wounds during the acute inflammatory phase, with caffeic acid phenethyl ester (CAPE) and other active compounds likely responsible for this effect [2,55]. CAPE, an active component of propolis, can reduce histamine release and the production of inflammatory cytokines in healing tissue [56,57].

Flavonoids and triterpenoids were reported [58] to promote wound healing, primarily due to their astringent and antimicrobial properties, which contribute to wound contraction and an increased rate of epithelization. Additionally, chrysin and kaempferol are major anti-allergic components of propolis [59]. Chrysin is able to decrease the gene expression of pro-inflammatory cytokines, such as TNF-α, IL-1β, IL-4, and IL-6, in mast cells through an NF-κB and caspase-1-dependent mechanism [59,60] (117, 118). It also reduces intracellular calcium levels in activated mast cells and suppresses calcium influx from outside the cells [59].

Phenolic compounds, particularly flavonoids, were reported to have wound healing activities due to their anti-inflammatory, angiogenic, re-epithelialization, and antioxidant effects [61,62]. Using GC–MS, flavonoids such as pinocembrin, pinostrobin chalcone, chrysin, and apigenin were identified as major and common constituents of Korean propolis in this study, and they might be responsible for the wound healing activities. Mazzotta et al. [63] reported pinocembrin and its 7-linolenoyl derivative were found to be innovative wound healing agents. Chrysin–curcumin-loaded nanofibers have anti-inflammatory properties in several stages of the wound healing process [64]. Lopez-Jornet et al. [65] reported topical applications of apigenin and PLX, showing faster re-epithelialization in the 7-day study period. Similarly, Daniela Balderas [50] reported the presence of flavonoids such as naringin, naringenin, kaempferol, quercetin, acacetin, luteolin, pinocembrin, and chrysin using HPLC–DAD and HPLC–MS analysis from Mexican propolis, and these compounds were reported to have anti-inflammatory, antioxidant, and antibacterial properties [66,67,68,69,70,71,72,73,74]. Further studies are needed to isolate and quantify the individual active principle(s) responsible for the wound healing properties of these investigated Korean propolis samples.

### 2.7. Anti-Inflammatory Activity

Excessive inflammatory mediators, such as tumor necrosis factor-α (TNF-α), interleukin-1β (IL-1β), interleukin-6 (IL-6), nitric oxide (NO), prostaglandin E_2_ (PGE_2_), inducible nitric oxide synthase (iNOS), and cyclooxygenase-2 (COX-2) have been linked to pathophysiological events and chronic inflammatory diseases. Therefore, controlling excessive inflammatory responses is important for the prevention of chronic inflammatory diseases. The anti-inflammatory effect of propolis extracts was evaluated by the method of NO production and expression levels of inflammatory mediators. As shown in Figure 6, the inhibitory effect on NO production was significantly highest (*p* < 0.05) in Gongju mountain propolis extracts among the ground propolis extracts compared with the other two samples. However, the inhibitory effect on the expression of iNOS, IL-1β, and IL-6 was similar in the propolis extracts of Gongju mountain and Gongju field propolis extracts.

The principal compounds of propolis samples such as pinocembrin and apigenin identified in this study were also previously reported from Turkish propolis. These compounds demonstrated the highest anti-inflammatory and antioxidant activities [75]. Based on the previous literature information, chrysin showed anti-inflammatory activity. One of the reports showed chrysin attenuated immunoglobulin E-mediated histamine release and inhibited TNFα, IL-1, IL-4, and IL-6 in the mast cell line RBL-2H3 [59,60,76]. Bae et al. also reported that chrysin inhibited the nuclear localization and transcriptional activation of NF-κB [60], and the mechanism underlying the anti-inflammatory activity of the compound was also reported [77]. 

Chrysin and its derivatives were also reported to suppress COX-2 and iNOS. Nitrate production activated by lipopolysaccharide (LPS) was suppressed by the treatment of cultured *Raw*264.7 cells with chrysin, 5-hydroxy-7-methoxyflavo, and 5,7-diacetylflavone [78]. Bruno et al. reported the other flavonoid present in Korean propolis apigenin was reported to have a potential therapeutic value in the treatment of various diseases such as cancer, diabetes, and cardiovascular and neurological disorders [79]. Ginwala reviewed that apigenin could be a promising candidate for the treatment of various inflammatory disorders, pending further investigation [80]. Lin et al. [81] reported pinostrobin chalcone isolated from rhizomes of *Alpinia pricei* showed strong anti-inflammatory effects. Therefore, the anti-inflammatory property of the EEP-K samples we studied may be attributed by major flavonoids such as chrysin and pinostrobin chalcone and related compounds.

## 3. Materials and Methods

### 3.1. Cell Culture Media and Chemicals

Dulbecco’s modified Eagle medium (DMEM) and lipopolysaccharide (LPS) (*Escherichia coli* 055:B5) were bought from Gibco Inc. (Grand Island, NY, USA) and from Sigma–Aldrich (St. Louis, MO, USA), respectively. We purchased SB203580 and PD98059 from Calbiochem (San Diego, CA, USA). We bought SB216763 and N-Acetyl Cysteine (NAC) from Sigma-Aldrich. Antibodies against inducible nitric oxide synthase (iNOS). and activating transcription factor 2 (ATF2) were obtained from Santa Cruz Biotechnology, Inc. (Santa Cruz, CA, USA), and those against inhibitor of nuclear factor kappa B (IκBα), *Transcription factor* (p65), extracellular-signal-regulated protein kinases 1 and 2 (ERK1/2), phospho-ERK1/2 (Thr202/Tyr204), and Actin were purchased from Cell Signaling (Bervely, MA, USA).

### 3.2. Sample Collection

Korean propolis produced by *Apis mellifera* were collected from Andong University, Beelab, Andong, and Gongju mountain and Gongju field in spring 2021.

### 3.3. Extraction

Extraction of propolis samples was carried out following the method of Uzel et al. [33], with some modification. The samples of Korean propolis produced by *Apis mellifera* was collected from propolis traps, which were frozen at −20 °C. The samples were then powdered using an electrical grinder. The ground propolis samples (50 g) were extracted in 95% *v*/*v* ethyl alcohol, in tightly closed bottles, for 3 days in an incubator at 37 °C, under occasional shaking in the dark. To remove waxes and less soluble substances, the suspensions were subsequently frozen at −20 °C for 24 h and then filtered. The freezing-filtration cycle was repeated three times for 2 weeks. The final filtrates of propolis were concentrated under reduced pressure at 40 °C to yield EEP. The percentage yields of the samples (*w*/*w*) were calculated and stored at 4 °C until analysis.

### 3.4. Thin-Layer Chromatography (TLC) Analysis

TLC analysis of ethanolic extracts of Korean propolis samples were carried out on silica gel 60 using solvent system petroleum ether/ethyl acetate 7:3 [35]. Visualization of the spots were obtained by UV light at 254 and 365 nm using a UV visualization chamber (Spectroline^R^, Model CM-10 Fluorescence Analysis Cabinet, Spectronics Corporation, Westbury, NY, USA).

### 3.5. Derivatization of EEP-K

EEP-K samples were derivatized based on the method reported by Musadji and Geffroy-Rodier [82]. Methanol (50 μL) and 200 μL of 12.5% (*w*/*v*) BF_3_-methanol were added to the vials containing the EEP-K sample (20 mg). After tightly capping, the vials were heated for 30 min at 70 °C. Then, after addition of pure water (100 μL) followed by capping, the vials were vigorously shaken, and then 150 μL of dichloromethane was added and then transferred to a separatory funnel. The dichloromethane layer containing the esterified fraction was obtained, concentrated, and analyzed by GC–MS.

### 3.6. GC–MS Condition

GC–MS analysis was performed using a GC (7890B, Agilent Technologies, Santa Clara, CA, USA) coupled with an MS (5977A Network, Agilent Technologies). The GC had an HP 5MS column (non-polar column, Agilent Technologies) and was 30 m × 250 μm in internal diameter (i.d.) and 0.25 μm in film thickness. The carrier gas was helium flowing at a rate of 1 mL/min. The injector temperature was 230 °C, and the injection mode was a split mode with split ratio 10:1. The initial oven temperature was 40 °C held for 5 min. It was raised to 250 °C at 6 °C/min and held at this temperature for 20 min. The total run time was 60 min. Mass spectra were recorded in EI mode at 70 eV, scanning the 50–650 *m*/*z* range. The identification of compounds was performed by comparing the mass spectra of the compounds with those in the database of NIST11 (National Institute of Standards and Technology, Gaithersburg, MD, USA) and by comparison with literature reports. Relative amounts of detected compounds were calculated based on the peak areas of the total ion chromatograms (TICs). Additionally, the chemical composition of propolis ethanolic extracts was analyzed based on the retention index (RI), which was calculated using a series of n-alkanes (C_7_–C_40_) [83].

### 3.7. Evaluation of DPPH Radical Scavenging Activity

The antioxidant activity of the propolis extracts was evaluated by DPPH radical scavenging assay. Reaction mixtures with 40 µL of propolis extract in DMSO and 760 µL of 300 µM DPPH ethanol solution in a micro tube were incubated at 37 °C for 30 min, and absorbance was measured at 515 nm based on increasing concentrations of the extracts. The DPPH quenching ability was calculated from the log-dose inhibition curve. The samples were analyzed in triplicates.

### 3.8. Cell Culture and Treatment

The mouse macrophage cell line RAW264.7 was obtained from Korean Cell Line Bank (Seoul, Republic of Korea) and cultured in DMEM supplemented with 10% fetal bovine serum (FBS), 100 U/mL penicillin, and 100 μg/mL streptomycin at 37 °C and 5% CO_2_ humidified atmosphere. The solvent used to dissolve propolis extracts was dimethyl sulfoxide (DMSO). After dissolving the extracts in the solvent, cell treatment followed. DMSO was used as a vehicle, and the final DMSO concentration did not exceed 0.1% (*v*/*v*).

### 3.9. Measurement of Nitric Oxide (NO) Production

Inhibition of propolis extracts on the formation of NO in LPS-stimulated RAW264.7 cells was evaluated using a Griess assay. RAW264.7 cells were inoculated in a 12-well plate for 24 h. Cells were first treated with propolis extracts for 2 h followed by co-treatment with LPS (1 μg/mL) for another 18 h. Then, the media (200 μL) were mixed with equal amount of Griess reagent (1% sulfanilamide and 0.1% N-1-(naphthyl) ethylenediamine dil. HCl in 2.5% phosphoric acid). The mixture was then incubated for 5 min more at 23 °C before the absorbance measurement at 540 nm.

### 3.10. Reverse Transcription Polymerase Chain Reaction (RT-PCR)

cDNA was synthesized using a Verso cDNA Kit (Thermo Scientific, Pittsburgh, PA, USA) from 1 μg of total RNA extracted from RAW264.7 cells using a RNeasy Mini Kit (Qiagen, Valencia, CA, USA). Then, PCR was performed using a PCR Master Mix Kit (Promega, Madison, WI, USA), and the sequences of the primers such as Inducible nitric oxide synthase (iNOS), Cyclooxygenase-2 (COX-2), Interleukin-1β (IL-1β), Interleukin 6 (IL-6), tumor necrosis factor-alpha (TNF-α), and glyceraldehyde-3-phosphate dehydrogenase (GAPDH) were as follows: iNOS: forward 5′-TTGTGCATCGACCTAGGCTGGAA-3′ and reverse 5′-GACCTTTCGCATTAGCATGGAAGC-3′, COX-2: forward 5′-GTACTGGCTCATGCTGGACGA-3′ and reverse 5′-CACCATACACTGCCAGGTCAGCAA-3′, IL-1β: forward 5′-GGCAGGCAGTATCACTCATT-3′ and reverse 5′-CCCAAGGCCACAGGTATTT-3′, IL-6: forward 5′-GAGGATACCACTCCCAACAGACC-3′ and reverse 5′-AAGTGCATCATCGTTGTTCATACA-3′, TNF-α: forward 5ʹ-TGGAACTGGCAGAAGAGGCA-3ʹ and reverse 5ʹ-TGCTCCTCCACTTGGTGGTT-3ʹ, GAPDH: forward 5′-GGACTGTGGTCATGAGCCCTTCCA-3′ and reverse 5′-ACTCACGGCAAATTCAACGGCAC-3′. The PCR results were visualized using agarose gel electrophoresis.

### 3.11. Experimental Animals

Healthy adult Swiss albino mice of both sexes, aged 6–8 weeks and weighing 26–36 g, were obtained from the animal house facility at the School of Pharmacy, Addis Ababa University (AAU). The animals were housed in cages under standard environmental conditions, including a 12 h light/dark cycle, with free access to standard laboratory pellets and clean drinking water at all times. Prior to the start of the experiments, the mice were allowed to acclimatize to the working environment for one week. The study was conducted in compliance with the guidelines established by the National Institutes of Health for the care and use of laboratory [84] animals.

### 3.12. Ointment Formulation

Based on preliminary tests conducted over the first ten days on 12 randomly selected mice, both the sticky crude form of propolis and its formulated ointment were tested to determine the appropriate dosage form and ingredient concentration. The crude propolis was tested at doses of 1 g, 2 g, and 4 g, corresponding to ointment formulations of 2.5%, 5%, and 10%, respectively. The ointment dosage form was found to be more effective at concentrations of 5% and 10% compared to the sticky crude extracts. Both simple and medicated ointments containing the ethanolic extract of propolis were prepared according to the standards outlined in the British Pharmacopoeia. A homogeneous 100 g simple ointment base was prepared by combining 5 g of hard paraffin, 5 g of cetylstearyl alcohol, 85 g of white soft paraffin, and 5 g of wool fat. The ingredients were added and melted in a water bath in descending order of their melting points. Ingredients used for the formulation of simple ointment with its formula were listed in Table 7.

Smooth medicated ointments of uniform consistency were prepared by incorporating 4 g of the crude extract into 36 g of the simple ointment base and 2 g of the crude extract into 38 g of the simple ointment for each crude propolis to yield 40 g of medicated ointment. By levigation, both the simple ointment and the stick of crude propolis were prepared on an ointment slab.

### 3.13. Acute Dermal Toxicity Test

An acute dermal toxicity test was conducted according to the OECD guideline number 404 [85]. Three adult female Swiss albino mice with normal skin texture were individually housed and allowed to acclimate to the laboratory conditions for 5 days prior to the start of the test. Before 24 h of conducting the study, the dorsal area of the trunk of the mice was shaved after administering anesthesia with ketamine at a dose of 50 mg/kg and diazepam 5 mg/kg intraperitoneally. A single dose of 2000 mg/kg (the maximum limit dose) of a 10% *w*/*w* crude extract of propolis was applied uniformly to the entire shaved site of one mouse and observed with special attention for the first 2 h. A confirmatory test was then performed on the other two mice after 24 h of observing the initial test. After removing the residual ointment, the mice were monitored daily for 14 days to observe the development of any adverse skin reactions, such as edema, tremors, convulsions, erythema, diarrhea, salivation, lethargy, and coma.

### 3.14. Grouping and Dosing of Animals

Three different crude extracts (Andong ethanolic propolis extract, Gongju field ethanolic propolis extract, and Gongju mountain ethanolic propolis extract) were evaluated using three types of wound models: the excision wound model, the incision wound model, and the dead space model. For excision wound models, animals were randomly separated into eight groups (each group containing six mice). Groups I, II, III, IV, V, VI, VII, and VIII were treated with simple ointment, 5% *w*/*w* Andong propolis (AP), 10% *w*/*w* (AP), 5% *w*/*w* Gongju field propolis (GFP), 10% *w*/*w* GFP, 5% *w*/*w* Gongju mountain propolis (GMP), 10% *w*/*w* GMP, and 0.2% *w*/*w* the standard drug, Nitrofurazone, respectively. Unlike the excision wound healing model, the incision wound healing model has one extra group (IX), which was left untreated and used as an untreated control.

### 3.15. Wound Healing Activity Tests

#### 3.15.1. Excision Wound Healing Models

##### Measurement of Wound Contraction

The wound healing rate is the rate of change in the wound surface area, which is an important metric used to evaluate the progress of wound healing. In this study, a progress of excision wound healing was assessed by measuring the surface area of the wound, including its length and width, using both a ruler as well as imaging techniques with a phone camera for wound contraction capacity, hydroxyproline contents, and recording epithelialization periods [86]. Prior to surgically creating the excision wound, the mice were administered ketamine at a dosage of 50 mg/kg and diazepam at a dosage of 5 mg/kg of body weight intraperitoneally [87]. Then, hair remover cream was applied on the back of animals, and their fur was shaved with a shaving device. After the animals’ fur was shaved, a circular wound area was meticulously created using sterile forceps and scissors, and the wound created was marked to measure the base area of the wound as shown in Figure 7. The wound was gently cleaned using a cotton swab soaked in a sterile saline solution to stop any bleeding and achieve hemostasis. The wound contraction rate was evaluated by measuring the wound margin area with a transparency sheet and permanent marker. The animals were observed for wound closure, and measurements were taken on the 2nd, 4th, 6th, 8th, 10th, 12th, 14th, 16th, 18th, and 20th days post-wounding.

##### Epithelialization Period Measurement

The epithelialization period was determined by counting the number of days needed for the dead tissue remnants to detach completely, leaving no raw wound behind [87].

#### 3.15.2. Incision Wound Model

Similar to that of the excision wound model, after anesthetizing the mice and their dorsal fur were shaved, a straight line was marked at a distance of 1 cm from the paravertebral region. Then, a 4 cm long longitudinal paravertebral incision was made through the skin and subcutaneous tissue. The parted skin was sutured 1 cm apart using a surgical thread with a curved needle on what was considered day 0. After 24 h of wound creation (on day 1), animals were treated as described under grouping and dosing in Section 3.13, with topical formulation of simple ointment, medicated ointment, and standard drug daily for nine days, leaving out the last group without applying any of the interventions. The sutures were removed on day 8 post-incision, and tensile strength was measured on the 10th post-wounding day using continuous water flow technique [86].

##### Measurement of Tensile Strength

On the 10th post-wounding day, mice were anesthetized and positioned on a working table. Two forceps were applied 1 cm from the healed tissue on either side of the incision. One forcep was stabilized, while the other was attached to a lightweight plastic bag via a pulley (Figure 8). Water flowed into the bag through an IV line, gradually increasing the weight and pulling the wound edges apart. Once the wound opened, water flow stopped, and the collected water volume was measured as an indirect indicator of breaking strength in grams [88].

The percent strength of the extracts, reference drugs, and ointments was then calculated using the specific formulas given below.
$$TS\;of\;extact(\%)=\frac{TS(extact)-TS(simple\;ointment)}{TS(simple\;ointment)}\times 100$$
$$TS\;of\;reference(\%)=\frac{TS(reference)-TS(simple\;ointment)}{TS(simple\;ointment)}\times 100$$
$$TS\;of\;simple\;ointment(\%)=\frac{TS(simple\;ointment)-TS(left\;untreated)}{TS(left\;untreated)}\times 100$$

#### 3.15.3. Dead Space Model

##### Determination of Hydroxyproline Contents

To determine standard hydroxyproline, the Neumann and Logan technique was employed [89]. First, 0.05 g of hydroxyproline was dissolved in 480 mL of water and 20 mL of concentrated hydrochloric acid to create a 100 µg/mL solution. One milliliter of this solution was further diluted to produce concentrations of 5, 10, and 15 µg/mL, with triplicate solutions prepared for each concentration along with a blank solution. In each tube, 1 mL of 0.05 M copper sulfate was added, followed by 1 mL of 2.5 N sodium hydroxide, and the contents were gently mixed by swirling. The tubes were placed in a water bath at 40 °C for 3–5 min, after which 1 mL of 6% hydrogen peroxide was added, mixing the contents before proceeding to the next tube. The tubes were left in the bath for 10 min, occasionally swirling them. After cooling with tap water, 4 mL of 3N sulfuric acid and 2 mL of 5% p-dimethylaminobenzaldehyde solution were added, with swirling after each addition. The tubes were capped and kept in a water bath at 70 °C for 16 min, then cooled and mixed, and their absorbance was measured at a wavelength of 572 nm.

The same procedure as described above was followed to create an excisional wound for measuring hydroxyproline content of treated groups by extracts. After 10 days of treatment with the formulations on the circular wound created in the excision model, each animal in the corresponding group was euthanized on the 11th day using a high dose of anesthesia. The tissues were then extracted from the mice with a surgical blade and dried in an oven at 60 °C for 12 h until reaching a constant weight. Following this, the dried tissue was hydrolyzed with 6 N hydrochloric acid for 24 h at 110 °C in sealed glass tubes. The hydrolysates were neutralized to pH 7.0. One milliliter of the supernatant from each sample was treated in the same manner as the standard hydroxyproline, and its absorbance was measured at a wavelength of 572 nm using a UV spectrophotometer [90]. The hydroxyproline content in each sample solution was calculated using the equation derived from the calibration curve (Appendix A).

### 3.16. Statistical Analysis

The experimental results of the study were presented as mean ± SEM (standard error mean). Data were entered, coded, verified, and analyzed using IBM SPSS version 27 software. Statistical significance was assessed using one-way ANOVA, followed by the Tukey post hoc test, with *p* < 0.05 considered statistically significant. The percentage of wound contraction was calculated based on the original wound area (mm^2^) from day zero.

### 3.17. Ethical Considerations

Before experimenting with the animal models, the study protocol was ethically approved by the Ethical Review Committee of the School of Pharmacy, College of Health Sciences, Addis Ababa University. All the experimental procedures were conducted abiding by the International Guidelines for the Use and Care of Laboratory Animals.

## 4. Conclusions

We have identified flavonoids such as pinocembrin, chrysin, and apigenin as major and common constituents of the three Korean propolis ethanolic extract samples analyzed. The propolis samples investigated showed some difference in their chemical compositions, wound healing, and antioxidant and anti-inflammatory activities. The botanical origin of the propolis from different apiaries might be different, which results in the propolis sample having different biological activities based on their secondary metabolites. Although our in vivo investigations demonstrated the beneficial roles of propolis, it is important to note that the composition of propolis varies between different propolis samples. Therefore, the characterization of the various propolis products is essential, and future studies comparing a wide range of propolis samples available in South Korea are recommended. Even though more studies are required, the highest wound healing and anti-inflammatory activities observed of Gongju mountain propolis might be due to its flavonoids and pinostrobin chalcone. Even though we identified the composition of Korean propolis ethanolic extracts, the compounds identified were not quantified in the present study. Our in vivo acute dermal toxicity study on rats treated with ethanolic extracts of propolis showed no signs of toxicity, providing a green light for further clinical studies. To standardize the development and application of propolis, ensuring reproducibility as a nutraceutical and/or pharmaceutical product, we recommend that future studies include the identification of bee species, geographical sources, and types of extracts. Further research into clinical studies and determining appropriate dosage in wound healing is also needed.

## Figures and Tables

**Figure 1 ijms-25-11352-f001:**
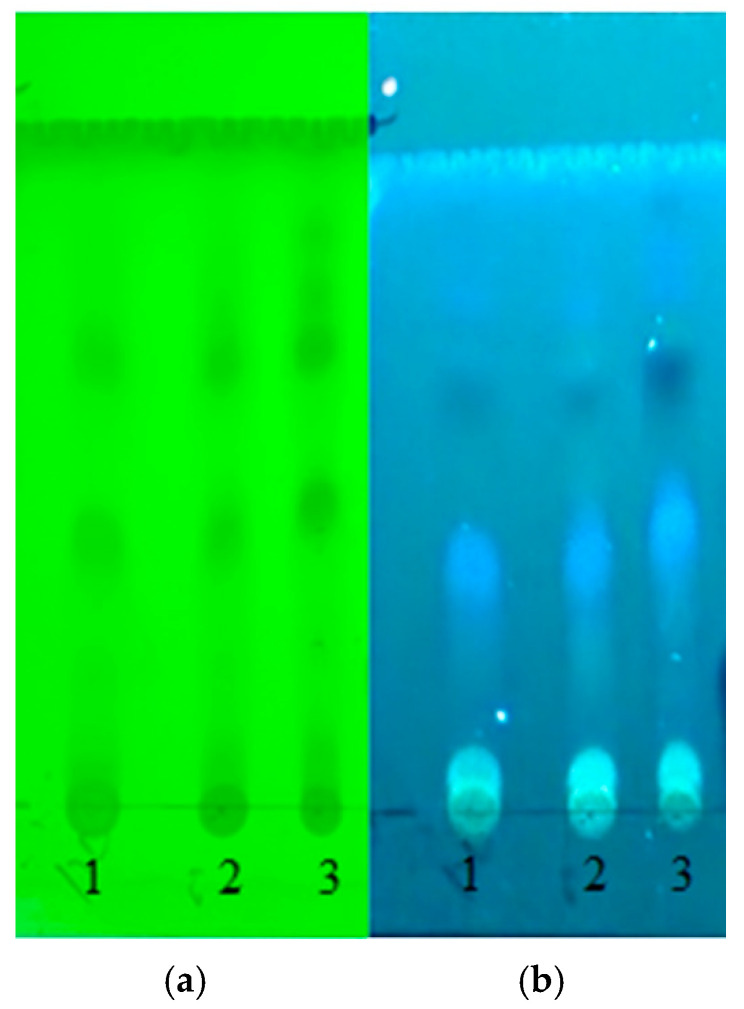
TLC spots under UV 254 nm (**a**) and 365 nm (**b**) of Andong (1), Gongju field (2), and Gongju mountain (3) EEP from left to right. TLC solvent: petroleum ether/EtOAc (7:3).

**Figure 2 ijms-25-11352-f002:**
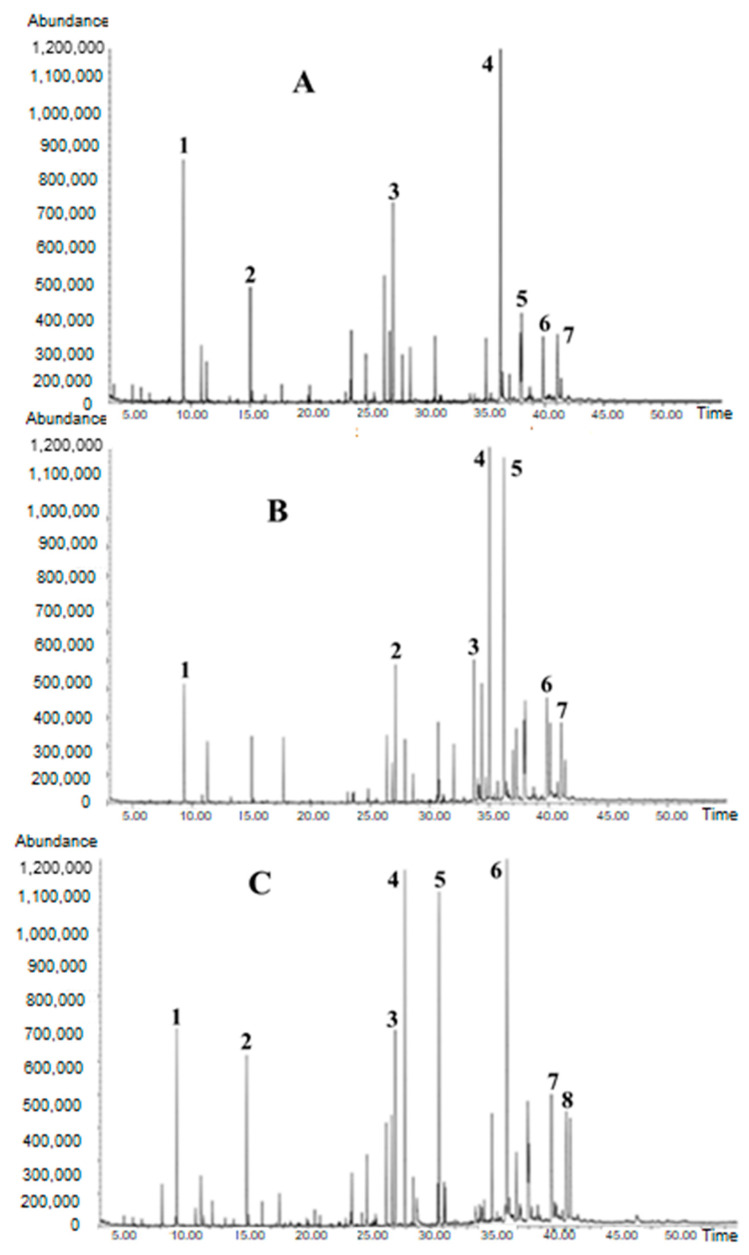
Gas chromatogram of EEP-K (**A**) collected from Andong. Major compounds: **1**. Benzyl alcohol (9.79%); **2**. 9-Methoxybicyclo[6.1.0]nona-2,4,6-triene (4.42%); **3**. Methyl 3,4-dimethoxycinnamate (7.76%); **4**. Pinocembrin (17.707%); **5**. 5-Hydroxy-7-methoxy flavone (5.44%); **6**. Chrysin (5.20%); **7**. Apigenin (5.84%). (**B**) Collected from Gongju mountain. Major compounds: **1**. Benzyl alcohol (3.399%); **2**. Methyl 3,4-dimethoxycinnamate (3.903%); **3**. Acridin-9-amine, 1,2,3,4-tetrahydro-5,8-dimethyl- (5.538%); **4**. Pinostrobin chalcone (10.681%); **5**. Pinocembrin (14.094%); **6**. Chrysin (6.752%); **7**. Apigenin (5.30%). (**C**) Collected from Gongju field. Major compounds: **1**. Benzyl alcohol (4.33%); **2**. 9-Methoxybicyclo[6.1.0]nona-2,4,6-triene (3.673%); **3**. Methyl 3,4-dimethoxycinnamate (4.409%); **4**. Pentadecanoic acid, 14-methyl-, methyl ester (7.559%); **5**. 8-Octadecenoic acid, methyl ester (7.823%); **6**. Pinocembrin (12.00%); **7**. Chrysin (6.65%); **8**. Apigenin (5.79%).

**Figure 3 ijms-25-11352-f003:**
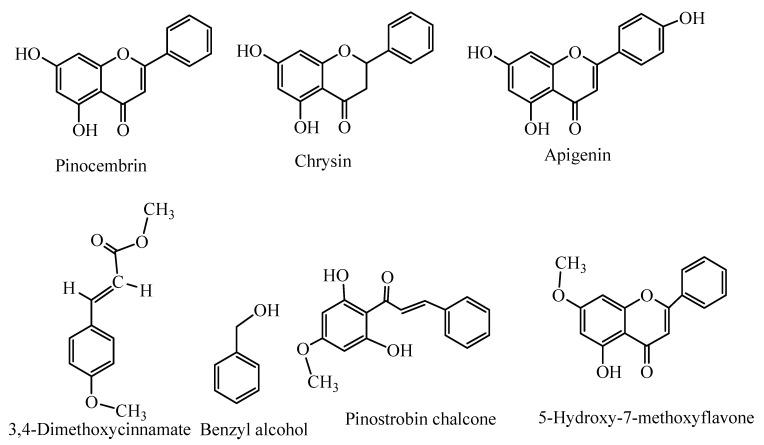
Structures of major compounds identified in the EEP-K.

**Figure 4 ijms-25-11352-f004:**
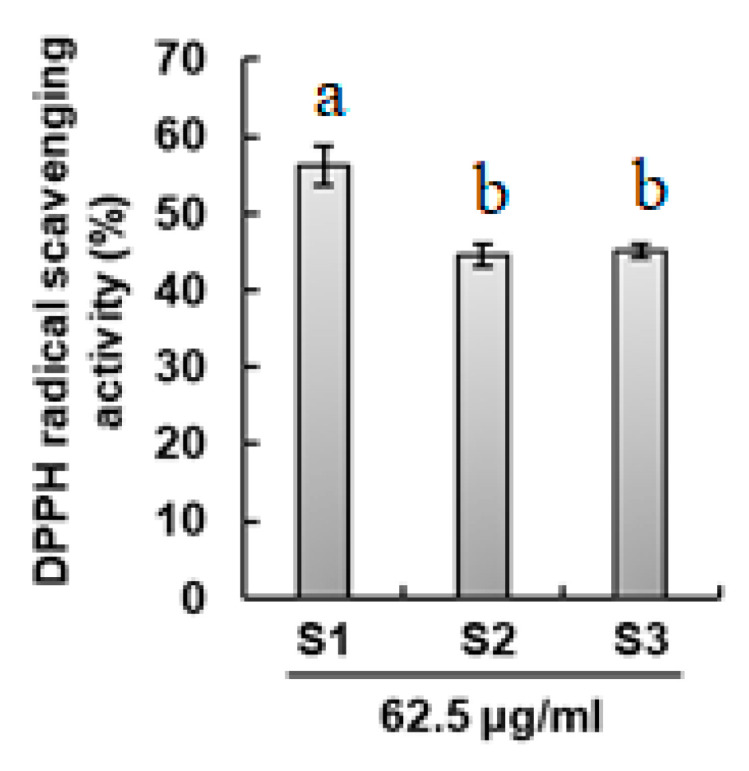
Antioxidant activities of propolis samples. S1: propolis collected from Andong; S2: propolis collected from Gongju mountain; S3: propolis collected from hives in fields of Gongju. Data are expressed as mean ± SD (*n* = 3). Values with no common letters are significantly different from each other (*p* ≤ 0.05).

**Figure 5 ijms-25-11352-f005:**
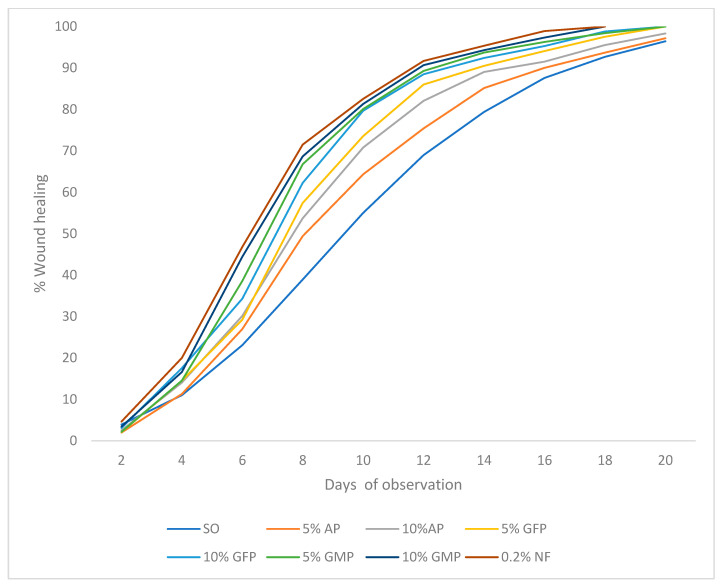
Percentage of wound closure of rats treated with ethanolic propolis extract in excision model. Values are expressed as mean ± SEM; *n* = 6. SO = simple ointment; 5% and 10% AP = 5% and 10% Andong ethanolic propolis extract, respectively; 5% and 10% GFP = 5% and 10% Gongju field ethanolic propolis extract, respectively; 5% and 10% GMP = 5% and 10% Gongju mountain ethanolic propolis extract and 0.2% NF = 0.2% nitrofurazone.

**Figure 6 ijms-25-11352-f006:**
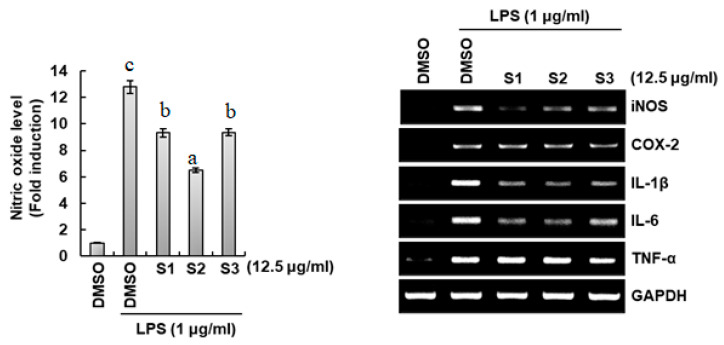
Anti-inflammatory activities of propolis samples. NO level (**left panel**) was measured using Griess assay, and the expression level (**right panel**) of iNOS, COX-2, IL-1β, IL-6, and TNF-α was measured using RT-PCR. S1: propolis collected from Andong, Beelab (A); S2: propolis collected from Gongju mountain (GM); S3: propolis collected from hives in fields of Gongju (GF). Data are mean ± SD. All samples were investigated in triplicate. Values with different letters showed significant difference (*p* ≤ 0.05).

**Figure 7 ijms-25-11352-f007:**
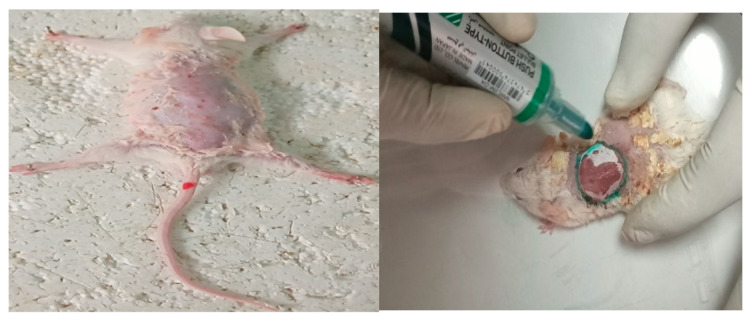
Photograph of animals’ fur removing and marking excision wound.

**Figure 8 ijms-25-11352-f008:**
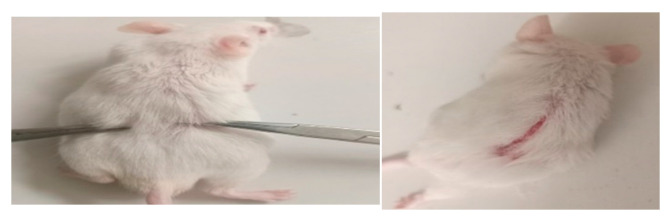
Tensile strength of incision wound by water flow technique.

**Table 1 ijms-25-11352-t001:** Percentage composition organic compounds in the EEP from Korea. A: Andong; GM: Gongju mountain; and GF: Gongju field propolis.

Compound ^i^	RT (min)	RI ^ii^	A	GM	GF
3-Methoxy-2-methyl-1-butene	3.433	600	0.3		
Benzene, chloro-	5.029	600	0.5		0.2
2-Methyl-2,4-dimethoxybutane	5.755	699	0.4		
3-Methoxy-3-methylbutanol	6.475	600	0.3		
Pentanoic acid, 4-oxo-, methyl ester	8.124	600			0.8
Butanedioic acid, dimethyl ester	9.244	600			0.2
Benzyl alcohol	9.333	599	9.8	3.4	4.3
Benzoic acid, methyl ester	10.86	701	2.1	0.2	0.3
Phenylethyl alcohol	11.295	599	1.8	1.9	1.2
3-Acetoxy-3-hydroxypropionic acid, methyl ester	11.512	599			0.3
Montanol	12.231	2100			0.5
Methyl salicylate	13.277	800	0.2		
9-Methoxybicyclo[6.1.0]nona-2,4,6-triene	15.028	999	4.4	1.9	
Benzenepropanoic acid, methyl ester	15.191	999	0.4		0.3
Decanoic acid, methyl ester	16.291	1100	0.3		0.5
2-Propenoic acid, 3-phenyl-, methyl ester	17.696	1001	0.8	2	0.8
γ-Selinene	19.916	1501	0.3		
1-Naphthalenol, 1,2,3,4-tetrahydro-, acetate	20.066	1000	0.7		
Dodecanoic acid, methyl ester	20.575	1299			0.4
Nonanedioic acid, dimethyl ester	20.996	1099			0.2
2-Quinolinecarbohydrazide	23.602	1000			1.4
α-Gurjunene	23.121	1499	0.4	0.3	
Methyl p-methoxycinnamate, cis	23.521	1100	0.9	0.4	0.3
2-(3,4-Dihydronaphthalen-1-yl)acetic acid	23.603	1200	2.8	0.4	
Methyl tetradecanoate	24.431	1501			0.3
2′,4′-Dihydroxyacetophenone oxime	24.763	801	0.3		
Methyl p-coumarate	24.831	1001	2.1	0.5	1.8
9H-Carbazole, 9-methyl-	25.524	1300			0.2
1,2-Dimethoxy-4-(1,2-dimethoxyethyl)benzene	25.531	1200	0.3		
Methyl ferulate	26.393	1101	4.7	2	2.4
2,4-Dimethoxycinnamic acid	26.848	1099	2.7	1.2	2.5
Methyl 3,4-dimethoxycinnamate	27.126	1099	7.8	3.9	4.4
Hexadecanoic acid, methyl ester	27.927	1700	1.7	1.8	7.5
Benzothiophene-3-carbonitrole, 4,5,6,7-tetrahydro-2-(4-methoxycarbonylbenzylidenamino)-	28.599	1999	2.0	0.8	1.1
9,12-Octadecadienoic acid (Z,Z)-, methyl ester	30.595	1899			0.9
9-Octadecenoic acid, methyl ester, (E)-	30.704	1899	2.7	2.5	7.82
11-Octadecenoic acid, methyl ester, (Z)-	30.792	1899		0.7	
Methyl stearate	31.111	1900	0.3	0.2	0.9
9-Hexadecenoic acid, methyl ester, (Z)-	31.213	1700	0.3		1.1
2-Propenoic acid, 3-phenyl-, 2-phenylethyl ester	32.027	1701		1.7	
Eicosane	33.663	2000	0.2		0.5
Acridin-9-amine, 1,2,3,4-tetrahydro-5,8-dimethyl-	33.724	1501		5.5	
Hexadecanoic acid, 14-methyl-, methyl ester	34.037	1699	0.3	0.7	0.6
1-Ethyl-3-(hexahydroazepin-2-ylidene)-2-indolinone	34.091	2199		0.5	
Methyl dehydroabietate	34.2	2099			0.7
1,2,4-Methenocyclobut[cd]inden-3(1H)-one, octahydro-	34.206	1099		0.7	
3-Heptanone, 5-hydroxy-1,7-diphenyl-	34.383	1900			0.5
Benzene, 3-hexenyl-	34.39	1200		3.6	
Pinostrobin chalcone	35.021	1600	2.9	10.7	2.7
Heneicosanoic acid, methyl ester	35.428	2201	0.3		0.2
Benzene, 1,2,3-trimethoxy-5-methyl-	35.734	999		0.6	
Androsta-1,4-diene-3,17-dione	36.107	1900			0.6
Pinocembrin	36.243	1500	17.7	14.1	12
Phenol, 3-pentadecyl-	36.44	2100	1.2	1.5	0.9
Docosanoic acid, methyl ester	37.003	2299	1.3	2.2	2.4
3-Methyldiphenylamine	37.282	1299		3.2	0.7
Benzenamine, 3,5-dimethoxy-	37.356	799		0.6	
2-Naphthalenecarbonitrile	37.363	1099			0.6
1-Methyl-6-oxo-1,6-dihydro-3-pyridinecarboxylic acid	37.934	700	4.4		4.8
2,4,5-Trihydroxyphenyl-p-chlorobenzylketone	37.94	1400		3.9	
5-hydroxy-7-methoxyflavone	38.035	1600	5.4	5	3.3
4H-3,1-Benzoxazin-4-one, 6,7-dimethoxy-2-(4-methoxyphenoxymethyl-	38.252	1800			0.5
2H-1-Benzopyran, 2,2-diphenyl-	38.64	2099	0.3		
Coumaran-5-ol-3-one, 2-[4-hydroxy-3-methoxybenzylidene]-	38.741	1599	0.9		0.7
3,3-Diphenyl-1-indanone	38.748	2099		0.7	
Chrysin	39.862	1499	5.2	6.8	6.7
Sarothrin	40.14	999		3.9	1.2
Hexadecanoic acid, 3-hydroxy-, methyl ester	40.276	1700	0.5		1
[1,2,4]Triazolo[1,5-a]pyrimidine-6-carboxylic acid, 4,7-dihydro-7-imino-, ethyl ester	40.466	1000	0.5		
1(2H)-Naphthalenone, 3,4-dihydro-2-(1-naphthalenylmethylene)-	40.758	2100		1.1	0.7
Apigenin	41.077	1500	5.8	5.3	5.8
Tetracosanoic acid, methyl ester	41.389	2499	1.7	2.5	5.2
Total			99.9	98.9	94.9

^i^ Compound arranged in the order of elution from HP 5MS column. ^ii^ Retention index relative to n-alkanes (C7–C40).

**Table 2 ijms-25-11352-t002:** Effect of propolis on contraction of excision wound in mice.

	Wound Area in mm^2^									
Group	2	4	6	8	10	12	14	16	18	20
SO	312.24 ± 1.42	281.15 ± 2.18	243.32 ± 1.19	193.16 ± 3.46	142.23 ± 2.66	98.20 ± 6.27	65.24 ± 2.93	39.24 ± 1.18	23.22 ± 2.15	11.27 ± 0.68
5% AP	306.16 ± 3.24	277.13 ± 3.17	228.23 ± 6.24	158.14 ± 4.56 ^a1^	111.44 ± 3.18 ^a1^	76.86 ± 2.46 ^a1^	46.08 ± 2.86 ^a1^	31.14 ± 2.86	19.71 ± 1.28	8.80 ± 1.14
10% AP	308.60 ± 3.18	272.16 ± 3.72	221.23 ± 9.06	146.55 ± 3.88 ^a2^	92.44 ± 4.28 ^a2^	56.82 ± 2.13 ^a2^	34.19 ± 1.30 ^a1^	25.46 ± 1.63 ^a2^	14.22 ± 1.55	5.32 ± 1.46
5% GFP	302.86 ± 4.19	264.43 ± 6.58 ^a2^	218.88 ± 4.14 ^a1 b2^	131.77 ± 2.34 ^a2b2c1^	82.42 ± 9.80 ^a3b2c1^	42.63 ± 2.36 ^a2b1c1^	29.37 ± 1.21 ^a2b3c32^	18.33 ± 2.84 ^a3b3c3^	7.66 ± 3.45 ^a2b3c2^	0 ^a3b2c3^
10% GFP	298.66 ± 2.33	254.15 ± 3.5 ^a2^	203.54 ± 2.19 ^a1b2^	116.31 ± 9.13 ^a1b2c1^	62.69 ± 7.5 ^a3b2c3^	34.65 ± 3.36 ^a3b2c1^	23.43 ± 2.64 ^a3b3c3^	14.56 ± 0.96 ^a3b2c2^	3.66 ± 2.44 ^a3b3c3^	0 ^a3b3c3^
5% GMP	300.82 ± 1.66	263.22 ± 1.11 ^a2b2^	188.90 ± 4.75 ^a3b2c2^	102.18 ± 4.44 ^a3b1c2^	61.25 ± 2.26 ^a2b1c1^	32.98 ± 6.12 ^a3b1c1^	19.33 ± 1.67 ^a2b3c3^	11.46 ± 2.68 ^a3b3c3^	1.92 ± 0.98 ^a3b2c3^	0 ^a3b3c3^
10% GMP	296.16 ± 4.48	255.75 ± 1.88 ^a1b1^	170.44 ± 6.43 ^a1b3c3^	96.14 ± 8.82 ^a2b3c3^	57.32 ± 4.37 ^a3b3c3^	28.33 ± 3.64 ^a3b3c3^	17.55 ± 2.12 ^a3b3c3^	7.98 ± 1.78 ^a2b3c3^	0 ^a3b3c3^	-
0.2% NF	297.86 ± 2.16	249.90 ± 3.13 ^a2b2^	166.22 ± 1.52 ^a2b3c2^	89.08 ± 2.5 ^a2b3c3^	54.54 ± 2.66 ^a3b3c3^	26.21 ± 1.84 ^a3b3c3^	14.48 ± 2.64 ^a3b3c3^	2.31 ± 2.42 ^a3b3c3^	0 ^a3b3c3^	-

Data are expressed as mean ± SEM (*n* = 6); all superscripts indicate significant difference against control (simple ointment); ^a^ against control (SO), ^b^ against 5% AP and ^c^ against 10% AP; ^1^ *p* < 0.05, ^2^ *p* < 0.01, ^3^ *p* < 0.001; SO = simple ointment; NF = nitrofurazone; AP = Andong ethanolic propolis extract; GFP = Gongju field ethanolic propolis extract; GMP = Gongju mountain ethanolic propolis extract; numbers from 2 to 20 indicate the day on which contraction rate measurement was taken.

**Table 3 ijms-25-11352-t003:** Effect of ethanolic extracts of propolis in percentage of wound closure excision model in mice.

Wound Closure in %										
Group	2	4	6	8	10	12	14	16	18	20
SO	3.92	11.01	23.06	38.92	55.02	68.95	79.37	87.59	92.67	96.44
5% AP	2.00	11.3	26.95	49.39	64.33	75.4	85.15	90.03	93.69	97.18
10% AP	2.53	14.04	30.13	53.71	70.8	82.05	89.02	91.51	95.51	98.32
5% GFP	2.09	14.51	29.23	57.4	73.53	85.99	90.50	94.07	97.52	100.00
10% GFP	3.1	17.54	34.29	62.26	79.66	88.47	92.40	95.28	98.81	100.00
5% GMP	2.17	14.52	38.57	66.77	80.08	89.27	93.71	96.27	98.38	100.00
10% GMP	3.42	16.6	44.42	68.65	81.31	90.66	94.28	97.34	100.00	
0.2% NF	4.64	20	46.79	71.49	82.54	91.69	95.36	98.91	100.00	

NF = nitrofurazone; AP = Andong ethanolic propolis extract; GFP = Gongju field ethanolic propolis extract; GMP = Gongju mountain ethanolic propolis extract. Numbers from 2 to 20 indicate the day on which contraction rate measurement was taken.

**Table 4 ijms-25-11352-t004:** Effects of ethanolic extraction of propolis on wound epithelization period in mice.

Group	Epithelization Period (Days)
SO	19.15 ± 0.44
5% AP	18.73 ± 0.53
10% AP	18.32 ± 0.29
5% GFP	16.43 ± 0.12 ^a1^
10% GFP	16 ± 0.18 ^a2b2c1^
5% GMP	16.43 ± 0.22 ^a3b1^
10% GMP	15.11 ± 0.44 ^a3b3c2^
0.2% NF	14.92 ± 0.64 ^a3b3c2^

Data are expressed as mean ± SEM (*n* = 6); all superscripts indicate significant difference against control (simple ointment); against control and NF = nitrofurazone; ^a^ against control (SO), ^b^ against 5% AP and ^c^ against 10% AP; ^1^ *p* < 0.05, ^2^ *p* < 0.01, and ^3^ *p* < 0.001. AP = Andong propolis, GFP = Gongju field propolis, and GMP = Gongju mount propolis.

**Table 5 ijms-25-11352-t005:** Effect of ethanolic extraction of propolis on tensile strength of incision wound in mice.

Group	Tensile Strength (g)	% Tensile Strength
Untreated control	178.18 ± 6.55	-
SO	206.56 ± 7.75	21.54
5% AP	295.14 ± 3.54 ^a1^	42.89
10% AP	304.66 ± 9.12 ^a2^	47.49
5% GFP	311.14 ± 4.33 ^a3b2c3^	50.63
10% GFP	321.92 ± 8.7 ^a2b3c2^	55.85
5% GMP	334.26 ± 7.79 ^a3b3c3^	61.82
10% GMP	338.13 ± 3.48 ^a3b3c3^	63.82
0.2% NF	352.76 ± 5.66 ^a3b3c3^	70.75

Data are expressed as mean ± SEM (*n* = 6); all superscripts indicate significant difference against control (simple ointment); NF = nitrofurazone; ^a^ against control (SO), ^b^ against 5% AP and ^c^ against 10% AP; ^1^ *p* < 0.05, ^2^ *p* < 0.01, and ^3^ *p* < 0.001; SO = simple ointment; NF = nitrofurazone; AP = Andong ethanolic propolis extract; GFP = Gongju field ethanolic propolis extract; GMP = Gongju mountain ethanolic propolis extract.

**Table 6 ijms-25-11352-t006:** Hydroxyproline content of excision wounds treated by ointment formulated from ethanol extract of propolis.

Group	Hydroxyproline Content (µg/15 mg of Tissues)
SO	4.72 ± 0.16
5% AP	7.09 ± 0.25 ^a3^
10% AP	9.15 ± 0.18 ^a3^
5% GFP	7.99 ± 0.24 ^a3b1c2^
10% GFP	9.03 ± 0.63 ^a2b1c1^
5% GMP	7.81 ± 0.47 ^a2b3c2^
10% GMP	9.42 ± 0.39 ^a3b2c2^
0.2% NF	9.89 ± 0.41 ^a3b2c3^

Data are expressed as mean ± SEM (*n* = 3); all superscripts indicate significant difference against control (simple ointment); NF = nitrofurazone; ^a^ against control (SO), ^b^ against 5% AP and ^c^ against 10% AP; ^1^ *p* < 0.05, ^2^ *p* < 0.01, and ^3^ *p* < 0.001; SO = simple ointment; NF = nitrofurazone; AP = Andong ethanolic propolis extract; GFP = Gongju field ethanolic propolis extract; GMP = Gongju mountain ethanolic propolis extract.

**Table 7 ijms-25-11352-t007:** Simple ointment preparation formula.

Ingredients	Master Formula (in g)	Reduced Formula (in g)
Wool fat	50	5
Hard paraffin	50	5
Cetylstearyl alcohol	50	5
White soft paraffin	850	85
Total	1000	100

## Data Availability

The data that support the findings of this study are available on request from the corresponding author.

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
