# Peer review of "Wound Healing, Anti-Inflammatory and Anti-Oxidant Activities, and Chemical Composition of Korean Propolis from Different Sources"

_ijms, 2024, doi:10.3390/ijms252111352_

Round 1
Reviewer 1 Report (Previous Reviewer 2)
Comments and Suggestions for Authors
The authors explored the efficacy of propolis on scar and wound healing.
Quality of the ms has been improved. There are some minor problems. In particular, propolis effects on wound healing has been studied and described. So, the authors need to discuss this point.
Comments on the Quality of English Languagefew english errors
Author Response
Reviewer 1
The authors explored the efficacy of propolis on scar and wound healing.
Quality of the ms has been improved. There are some minor problems. In particular, propolis effects on wound healing has been studied and described. So, the authors need to discuss this point.
- We appreciate the reviewer's comment and have revised the manuscript accordingly. Additionally, we have expanded our discussion on the wound healing properties of propolis.
Reviewer 2 Report (New Reviewer)
Comments and Suggestions for Authors
The current study is interesting and introduces significant findings; however, some comments should be addressed as follows:
1. title: please add (Korean propolis from different sources).
2. Abstract: Please highlight that this is a comparative study between three kinds of propolis. Significant findings should be discussed in terms of biological activities. Line 24-25: please correct this sentence.
3. Introduction: it is too short. Please describe the properties of propolis from different sources using updated references in 2024. The novelty of this study should be justified and the aims should be clearly outlined.
4. Results and discussion: Please number the samples in Fig. 1 and illustrate why they indicate in the figure legend. Fig. 2: they should be Improved. Section 2.3.: why did you select concentration of 62.5 µg/ml. Fig. 5: please redraw it to avoid the overlapping, like dash. The image of rats throughout the duration of experiment should be presented. Fig. 5: please mention the figure legends what the DNA electrophoresis refers to.
5. Methods: line 353: Apis mellifera L. Please add a reference for DPPH assay. Please describe the sequence of primers using capital letters. You mentioned the equation for wound closure rate two times in the manuscript, please remove one of them.
6. Conclusion: it is too short. The limitation of this study and future perspectives should be discussed.
Author Response
The current study is interesting and introduces significant findings; however, some comments should be addressed as follows:
- title: please add (Korean propolis from different sources).
- The suggested correction was done.
- Abstract: Please highlight that this is a comparative study between three kinds of propolis. Significant findings should be discussed in terms of biological activities. Line 24-25: please correct this sentence.
- We thank the reviewer for their valuable comments and have incorporated the suggested corrections in the manuscript.
- Introduction: it is too short. Please describe the properties of propolis from different sources using updated references in 2024. The novelty of this study should be justified and the aims should be clearly outlined.
- We appreciate the reviewer for their valuable comment. We have added relevant information using related articles, including those published in 2024. The novelty of the work has been justified, and its aims have been clearly stated.
- Results and discussion:
Please number the samples in Fig. 1 and illustrate what they indicate in the figure legend.
- The samples in the TLC profile shown in Fig. 1 have been numbered.
Fig. 2: they should be Improved.
- We have improved the quality of Figure 2 to 300 dpi. However, to compare the GC peaks of the three EEP samples, we reduced the GC scale, which impacted the overall image quality.
Section 2.3.: why did you select concentration of 62.5 µg/ml.
- The reason for selecting the concentration of 62.5 μg/ml for the comparison of DPPH scavenging activity in this study is that significant differences of activity were observed among the samples at this concentration. Experiments were conducted using a range of concentrations from 25 μg/ml to 500 μg/ml. At higher concentrations, DPPH were high close to 100% in all samples. However, in 62.5 μg/ml concentration, clearer differentiation of antioxidant effects among the samples were noticed. Therefore, the data presented in this study focuses on the 62.5 μg/ml concentration to facilitate a more meaningful comparison of the samples' activities.
Fig. 5: please redraw it to avoid the overlapping, like dash.
- We appreciate the comment provided. We have redrawn Figure 5 by adjusting the scale and reducing the line width to minimize overlap. As a result, the overlap between the lines has been significantly reduced.
The image of rats throughout the duration of experiment should be presented.
- To reduce the number of figures in the manuscript, we have moved the image of the rats during the experiment to the supplementary information (Supplementary Figure S1). The effect of propolis on wound contraction in the excision wound model in rats is also presented in Table 2. If the number of figures in the manuscript is not an issue, we can relocate the image back to the main manuscript.
Fig. 5: please mention the figure legends.
- We included legends for Figure 5.
What the DNA electrophoresis refers to?
- Gel electrophoresis mentioned in the section 3.9, refers to the instrument used to separate charged molecules, like DNA, according to size.
- Methods: line 353: Apis mellifera
- We included the suggested correction.
Please add a reference for DPPH assay.
- We conducted the antioxidant activity following the method described by the following article.
Blois M.S. Antioxidant determinations by the use of a stable free radical. J. Agric. Food Chem. 1977,25,103-107.
Please describe the sequence of primers using capital letters.
- Thank you for comment. We changed described the sequence of primers using capital letters
You mentioned the equation for wound closure rate two times in the manuscript, please remove one of them.
- We removed equation for wound closure from section 14.1. Excision wound healing models
- Conclusion: it is too short. The limitation of this study and future perspectives should be discussed.
- We amended the conclusion section.
Round 2
Reviewer 2 Report (New Reviewer)
Comments and Suggestions for Authors
The authors addressed all comments carefully, but I have one comment (please correct the sentence lines 572–573).
Author Response
The authors addressed all comments carefully, but I have one comment (please correct the sentence lines 572–573).
==> Thank you. We corrected and checked one more time.
This manuscript is a resubmission of an earlier submission. The following is a list of the peer review reports and author responses from that submission.
Round 1
Reviewer 1 Report
Comments and Suggestions for Authors
In this manuscript, Dekebo et al. set out to investigate the chemical composition and biological activities of Korean propolis collected from different regions. They highlight the variation in chemical composition due to geographical factors and the lack of previous studies comparing propolis from different regions in South Korea. Overall, the manuscript presents valuable research on Korean propolis, its chemical composition, and biological activities.
Minor Comments:
- Line 13: what does flavour have to do with bio-cosmetics?
- Line 36: how does propolis “prevent wounds?”
- Some abbreviations (e.g., EEP-K and many in methods section) are introduced without prior definition.
- Line 207: 3.3 Extraction – this method paragraph is difficult to follow and should be rewritten for clarity. E.g., what does “the forementioned extraction procedure was done for 2 weeks” mean after mentioning the filtrate was frozen for 24 hours?
- Line 212: “feezed” what does this mean? Should it be “frozen””?
- Line 282: “Analysis of valued” what does this mean?
Comments on the Quality of English LanguageThe overall quality of English language needs improvement. Proofreading for grammatical errors, sentence structure, and word choice is required. Below are some examples (not exhaustive – please proofread entire manuscript):
- Line 13: incomplete, “the use of propolis as” ?
- Line 22: “showed highly differences” change to “substantial differences”
- Line 31: Add “the” before hive collected by honeybees"
- Line 35: “value since ancient times” change to “, valued since ancient times”
- Line 42: “which leads to increased interest” change to “which has led to increased interest”
- Line 52: This sentence is too long and run on. Break into smaller sentences.
- Line 172: “literature” not “literatur”
- Line 200: add “from” before “Sigma-Aldrich
- Line 205: change to “Koren propolis produced by Apis mellifera was collected from…”
- Line 209: “was frozen” change to “and were frozen”
- Line 284: “Turkey” should be “Tukey”
- Line 109: “collected Chihuahua” should be “collected from Chihuahua”
- Line 110: “had a highest yield” should be “a high yield” or “the highest yield”
- Line 142: This sentence is too long and run on. Break into smaller sentences.
Author Response
Comments and Suggestions for Authors
In this manuscript, Dekebo et al. set out to investigate the chemical composition and biological activities of Korean propolis collected from different regions. They highlight the variation in chemical composition due to geographical factors and the lack of previous studies comparing propolis from different regions in South Korea. Overall, the manuscript presents valuable research on Korean propolis, its chemical composition, and biological activities.
Minor Comments:
- Line 13: what does flavour have to do with bio-cosmetics?
Thank you for the correction. We changed flavor with aroma in the manuscript.
- Line 36: how does propolis “prevent wounds?”
Cao et al (2017) reported propolis ethanol extracts efficiently reduced the excessive accumulation of ROS, protecting skin cells from oxidative injury which explains that propolis is useful in wound healing. We included this information and reference in the manuscript.
- Some abbreviations (e.g., EEP-K and many in methods section) are introduced without prior definition.
We corrected as ethanol extract of Korean Propolis (EEKP) in the abstract.
- Line 207: 3.3 Extraction – this method paragraph is difficult to follow and should be rewritten for clarity. E.g., what does “the forementioned extraction procedure was done for 2 weeks” mean after mentioning the filtrate was frozen for 24 hours?
The correction was done in the manuscript based on the comments given.
- Line 212: “feezed” what does this mean? Should it be “frozen””?
We changed feezed to frozen in the manuscript.
- Line 282: “Analysis of valued” what does this mean?
It was corrected as follows in the manuscript. Statistical analysis was done with SPSS program.
Comments on the Quality of English Language
The overall quality of English language needs improvement. Proofreading for grammatical errors, sentence structure, and word choice is required. Below are some examples (not exhaustive – please proofread entire manuscript): Thank you very much for the comments. Language proofreading and correction were made from the consultation from a professional English consulting (proof attached).
- Line 13: incomplete, “the use of propolis as” ?
The correction was done in the manuscript.
- Line 22: “showed highly differences” change to “substantial differences”
The correction was done in the manuscript
- Line 31: Add “the” before hive collected by honeybees"
The correction was done in the manuscript
- Line 35: “value since ancient times” change to “, valued since ancient times”
The correction was done in the manuscript
- Line 42: “which leads to increased interest” change to “which has led to increased interest”
The correction was done in the manuscript.
- Line 52: This sentence is too long and run on. Break into smaller sentences.
The correction was done in the manuscript.
- Line 172: “literature” not “literatur”
The correction was incorporated in the manuscript.
- Line 200: add “from” before “Sigma-Aldrich
The correction was incorporated in the manuscript.
- Line 205: change to “Koren propolis produced by Apis mellifera was collected from…”
The correction was done in the manuscript.
- Line 209: “was frozen” change to “and were frozen”
The correction was done in the manuscript.
- Line 284: “Turkey” should be “Tukey”
The correction was done in the manuscript.
- Line 109: “collected Chihuahua” should be “collected from Chihuahua”
It was corrected in the manuscript.
- Line 110: “had a highest yield” should be “a high yield” or “the highest yield”
It was changed to the highest yield in the manuscript.
- Line 142: This sentence is too long and run on. Break into smaller sentences.
English writing was completely checked from the professional consulting. Thank you for the comments.
Reviewer 2 Report
Comments and Suggestions for Authors
The ms explores the composition, anti-oxidant and anti-inflammatory properties of some korean propolis. Although the topic is interesting, I can not propose the acceptance of the ms beacuse I do not find novelty or originality that IJMS needs. In fact, there are a lot of scientific literature exploring the anti-inflammatory and anti-oxidants properties of propolis, but the authors did not explore other biological positive role of propolis (eg. wound-healing properties). Moreover, no comparison with well-kwown propolis samples is proposed.
Comments on the Quality of English Languageminor english corrections
Author Response
Review 2
Yes Can be improved Must be improved Not applicable
Does the introduction provide sufficient background and include all relevant references?
( ) ( ) (x) ( )
Are all the cited references relevant to the research?
( ) ( ) (x) ( )
Is the research design appropriate?
( ) ( ) (x) ( )
Are the methods adequately described?
( ) ( ) (x) ( )
Are the results clearly presented?
( ) ( ) (x) ( )
Are the conclusions supported by the results?
( ) ( ) (x) ( )
Comments and Suggestions for Authors
The ms explores the composition, anti-oxidant and anti-inflammatory properties of some Korean propolis. Although the topic is interesting, I cannot propose the acceptance of the ms because I do not find novelty or originality that IJMS needs. In fact, there are a lot of scientific literature exploring the anti-inflammatory and anti-oxidants properties of propolis, but the authors did not explore other biological positive role of propolis (eg. wound-healing properties).
Even though there is a lot of literature information about anti-inflammatory and anti-oxidants properties of propolis, those data were not about the propolis samples we investigated from two regions in Korea. We have shown clearly propolis from different regions or even from different places in the same regions have some different chemical composition, which results in different antioxidant and anti-inflammatory activities. Therefore, the work conducted in this study filled those research gaps. We appreciate the reviewer 2 for suggesting to conduct wound healing effects of the propolis samples. But because of research facility issues, it is difficult for us to investigate those properties and include in this manuscript. We cited the research work by Cao et al (2017) which explains the mechanism underlying for the wound healing potential of propolis.
Moreover, no comparison with well-known propolis samples is proposed.
In the manuscript, we compared our results of chemical composition, anti-inflammatory activities and anti-oxidant activities of Korean propolis with well-known propolis samples. Thus the explicit comparison may be limited but relative comparison is possible.
Based on the reviewers’ comment, we had revised the manuscript more meaningful and scientifically sound. Thank you again for your valuable time for the comments.
Round 2
Reviewer 2 Report
Comments and Suggestions for Authors
The authors did not provide enough new input to suggest pubblication.
Comments on the Quality of English Languagenone
Author Response
September 13/2024
Point by Point Response Letter
Manuscript ID: ijms-2704071
Type of manuscript: Article
Title: Wound healing, anti-Inflammatory and anti-oxidant activities and chemical composition of Korean propolis
To: Assistant Editor,
Submitted to section: Bioactives and Nutraceuticals,
Subject: Point by point response to Editor and Reviewers comments
Dear Assistant Editor,
We thank the editor and the reviewers for their comprehensive review and constructive suggestions on manuscript with ID ijms-2704071. Please kindly find below our response to each point raised by the academic editor and reviewers. We addressed all comments raised and the manuscript will be now suitable for publication. Below are our point-by-point responses and accordingly the changes have been made in the manuscript and highlighted.
Sincerely,
On behalf of all authors,
Chuleui Jung (Prof.)
Response to Editor comments
(I) Please check that all references are relevant to the contents of the
manuscript.
Thank you for valuable comments. We have checked all references again for their relevance and updated them.
(II) Any revisions to the manuscript should be highlighted, such that any
changes can be easily reviewed by editors and reviewers.
We have highlighted all the changed made.
(III) Please provide a cover letter to explain, point by point, the details
of the revisions to the manuscript and your responses to the referees’
comments.
That was done.
(IV) If you found it impossible to address certain comments in the review
reports, please include an explanation in your appeal.
That was done.
(V) The revised version will be sent to the editors and reviewers.
Thank you for the information.
Response to Reviewers
Reviewer #1:
Comments and Suggestions for Authors
In this manuscript, Dekebo et al. set out to investigate the chemical composition and biological activities of Korean propolis collected from different regions. They highlight the variation in chemical composition due to geographical factors and the lack of previous studies comparing propolis from different regions in South Korea. Overall, the manuscript presents valuable research on Korean propolis, its chemical composition, and biological activities.
It is a pleasure to receive your valuable comments. In response, we revised the manuscript in accordance with the comments.
Minor Comments:
- Line 13: what does flavour have to do with bio-cosmetics?
Thank you for the comment. We replaced flavour with antioxidant.
- Line 36: how does propolis “prevent wounds?”
We corrected line 36 as proposlis was claimed to promotes tissue repair, accelerates wound contraction, shortens healing time, and reduces scar formation after wound healing.
- Some abbreviations (e.g., EEP-K and many in methods section) are introduced without prior definition.
We wrote full names of terms before using abbreviations. - Line 207: 3.3 Extraction – this method paragraph is difficult to follow and should be rewritten for clarity. E.g., what does “the forementioned extraction procedure was done for 2 weeks” mean after mentioning the filtrate was frozen for 24 hours?
Thank you for the valuable comments. The extraction section was rewritten.
- Line 212: “feezed” what does this mean? Should it be “frozen””?
It was corrected as frozen in the manuscript.
- Line 282: “Analysis of valued” what does this mean?
The correction was done in the manuscript.
- Comments on the Quality of English Language
The overall quality of English language needs improvement. Proofreading for grammatical errors, sentence structure, and word choice is required. Below are some examples (not exhaustive – please proofread entire manuscript):
The language of the manuscript was checked by native speaker and improved.
- Line 13: incomplete, “the use of propolis as” ?
The correction was done in the manuscript.
- Line 22: “showed highly differences” change to “substantial differences”
The correction was done in the manuscript.
- Line 31: Add “the” before hive collected by honeybees"
The correction was incorporated in the manuscript.
- Line 35: “value since ancient times” change to “, valued since ancient times”
The correction was done in the manuscript.
- Line 42: “which leads to increased interest” change to “which has led to increased interest”
The correction was incorporated in the manuscript.
- Line 52: This sentence is too long and run on. Break into smaller sentences.
The suggested correction was done in the manuscript and highlighted.
- Line 172: “literature” not “literatur”
The correction was done in the manuscript.
- Line 200: add “from” before “Sigma-Aldrich
The correction was incorporated in the manuscript.
- Line 205: change to “Koren propolis produced by Apis mellifera was collected from…”
The correction was done in the manuscript and highlighted.
- Line 209: “was frozen” change to “and were frozen”
The correction was added.
- Line 284: “Turkey” should be “Tukey”
The correction was done.
- Line 109: “collected Chihuahua” should be “collected from Chihuahua”
The correction was done.
- Line 110: “had a highest yield” should be “a high yield” or “the highest yield”
The correction was done in the manuscript and highlighted.
- Line 142: This sentence is too long and run on. Break into smaller sentences.
The suggested correction was done.
Review #2
Yes Can be improved Must be improved Not applicable
Does the introduction provide sufficient background and include all relevant references?
( ) ( ) (x) ( )
Are all the cited references relevant to the research?
( ) ( ) (x) ( )
Is the research design appropriate?
( ) ( ) (x) ( )
Are the methods adequately described?
( ) ( ) (x) ( )
Are the results clearly presented?
( ) ( ) (x) ( )
Are the conclusions supported by the results?
( ) ( ) (x) ( )
Comments and Suggestions for Authors
The ms explores the composition, anti-oxidant and anti-inflammatory properties of some korean propolis. Although the topic is interesting, I can not propose the acceptance of the ms because I do not find novelty or originality that IJMS needs. In fact, there are a lot of scientific literature exploring the anti-inflammatory and anti-oxidants properties of propolis, but the authors did not explore other biological positive role of propolis (eg. wound-healing properties). Moreover, no comparison with well-kwown propolis samples is proposed.
We appreciate the major correction suggested by Reviewer 2 and acknowledge the importance of evaluating the wound-healing properties. In response, we collaborated with researchers from the Department of Pharmacognosy and Pharmaceutical Chemistry at Addis Ababa University, who conducted in vivo tests on the ethanolic propolis extract using three wound models: the excision wound model, the incision wound model, and the dead space model. The results of this study have now been included in the manuscript.
